# The Impact of *Campylobacter*, *Salmonella*, and *Shigella* in Diarrheal Infections in Central Africa (1998–2022): A Systematic Review

**DOI:** 10.3390/ijerph21121635

**Published:** 2024-12-08

**Authors:** Ornella Zong Minko, Rolande Mabika Mabika, Rachel Moyen, Franck Mounioko, Léonce Fauster Ondjiangui, Jean Fabrice Yala

**Affiliations:** 1Bacteriology Laboratory, Medical Analysis Research Unit, Interdisciplinary Center for Medical Research of Franceville (CIRMF), Franceville BP 769, Gabon; ornellapacy@gmail.com (O.Z.M.); rolandemabika@gmail.com (R.M.M.); leoncefauster@gmail.com (L.F.O.); 2Molecular and Cellular Biology Laboratory, Microbiology Team (LABMC), Agrobiology Research Unit, Masuku University of Sciences and Techniques (USTM), Franceville BP 067, Gabon; 3Laboratory of Cellular and Molecular Biology, Sciences and Techniques Faculty, University Marien Ngouabi, Brazzaville BP 69, Congo; rmoyen@yahoo.fr; 4Vector Systems Ecology Unit, Interdisciplinary Center for Medical Research of Franceville (CIRMF), Franceville BP 769, Gabon; fmounioko@yahoo.fr

**Keywords:** Central Africa, *Campylobacter*, *Salmonella* or *Shigella*, diarrhea, prevalence

## Abstract

Background: Gastric diseases caused, in particular, by *Campylobacter*, non-typhoidal *Salmonella*, and *Shigella* resulting from food and/or water problems, are a disproportionately distributed burden in developing countries in Central Africa. The aim of this work was to compile a list of studies establishing the prevalence of the involvement of these bacterial genera in diarrheal syndromes in Central Africa from 1998 to 2022. Methods: The Preferred Reporting Articles for Systemic Reviews and Meta-Analyses, six (6) database (Pubmed, Google Scholar, Semantic Scholar, Freefullpdf, and Scinapse) were perused for research on the role of *Campylobacter*, *Salmonella* and *Shigella* diarrheal infections in humans and animals, in 9 country of Central Africa over from 1998 to 2022. Results: Seventeen articles were selected, including 16 on humans and one on animals. These data were recorded in 6 of the 9 countries of Central Africa, including Gabon (5), Angola (3), Cameroon (3), the Democratic Republic of Congo (3), Chad (2), and the Central African Republic (1). Mono-infections with *Salmonella* spp. were the most predominant (55.56%, n = 5/9), followed by an equal proportion of *Campylobacter* spp. and *Shigella* spp. with 44.44% (4/9), respectively and, co-infections with *Campylobacter/Salmonella* spp. and *Salmonella/Shigella* spp. with a prevalence of 11.11% (1/9) respectively. The most used diagnostic tool was conventional culture (82.35%) against 17.65% for PCR or real-time PCR. Conclusion: Despite the paucity of recorded data on the prevalence of diarrheal infections due to *Campylobacter* in this sub-region, it is crucial that scientific studies focus on the diagnosis and monitoring of this zoonotic bacterium. Also, improved diagnosis will necessarily involve the integration of molecular tools in the diagnosis of these diarrheic syndromes in both humans and animals.

## 1. Introduction

Food-borne and water-borne diseases are major public health problems worldwide, with significant morbidity and mortality affecting both developed and developing countries [1,2]. Nearly 600 million people worldwide have been affected by food-related diseases [3]. These figures are constantly rising, and the health news linked to the COVID-19 pandemic recently revealed that 3 billion people in Asia and Africa do not have access to a balanced diet [4]. Clearly, these diseases are favored by poor hygiene and difficult access to drinking water sources. Indeed, 28% of the population of sub-Saharan Africa defecates in the open air, and 23% use sanitary facilities that do not ensure hygienic separation of excreta from the immediate human environment [5]. Also, in 2015, nearly 2.5 million people worldwide did not have access to a clean, reliable water source or adequate sanitation facilities [2]. 2.5 billion people still lack access to adequate sanitation and clean water.

Although they are mainly caused by viruses, bacteria and parasites, foodborne and waterborne diseases generally manifest themselves through gastrointestinal symptoms [6]. These gastrointestinal symptoms are relatively common to all types of microorganisms that cause diarrhea [7]. Indeed, despite the fact that bacteria account for 50–60% of microorganisms isolated during acute diarrheal episodes, most studies in Black Africa seem to focus primarily on diarrheal infections caused by *Escherichia coli* Diarrheic, to the detriment of other genera such as *Campylobacter*, *Salmonella* spp., and *Shigella* spp. [8]. This high detection rate is due to the fact that *Escherichia coli*, a commensal enteropathogenic bacterium regularly involved in infectious human diarrhea, is found in 80% of the intestinal microbiota of humans and warm-blooded animals [9]. In addition, although *Campylobacter* and *Salmonella* infections are the leading causes of zoonotic food poisoning worldwide [9,10]. In 2017, the annual incidence of infections caused by toxins of these two bacterial genera ranged from 3.1 to 5 million in Canada [1].According to these authors, in Australia, this figure was as high as 5.4 million [1]. Nevertheless, the impact of *Campylobacter* and *Salmonella* on diarrheal is better studied in industrialized than in developing countries. In addition, the consumption of raw red meat, raw milk, unsanitary drinking water and direct contact with farm animals and the presence of *Campylobacter* in 80% of animal intestines have all been reported as major sources of human campylobacteriosis worldwide [11].

The estimated rate of *Campylobacter*-related diarrheal illness in the European Union (EU) is currently 45.2 cases per 100,000 people, with grilled chicken and fresh turkey being the main sources of infection. *Salmonella* infections, which have been linked to increased temperature, have been reported in 31.1 cases per 100,000 people [12,13]. While the 2012 ban on eating raw beef served in restaurants in Asia, including Japan, may have helped minimize *Salmonella* infections, the number of *Campylobacter* cases remains high and is estimated to have risen from 551 to 2643 between 2000 and 2018 [13].

In Africa, data on these infections remain relatively fragmented and country-specific. In most West African countries, the predominance of livestock and meat production guides research on *Campylobacter* infections in livestock sectors. According to a study by Audu et al., published in 2022, in Nigeria, the prevalence of *Campylobacter* is estimated at 9 and 28% in humans and animal faeces, respectively [14]. Salmonellosis is also widespread in that subregion, with a prevalence of 47.9% in Nigerian poultry farms [15]. In Addis Ababa, Ethiopia, in Eastern Africa, the study of Chala et al., published in 2021 conducted in a peri-urban area reported the prevalence of *Campylobacter* in humans (10.1%), cattle (18.5%), poultry (13.0%), sheep (13.3%), goats (7.1%) and water (10.5%). This study also reports a significant rate of detection of *Campylobacter* in domestic animals near households (42.4%) [16].

In North Africa, in the Casablanca-Sett region of Morocco, Es-Soucratti et al. (2020) detected *Campylobacter* in 73% of poultry isolates, with 102 samples out of 140 testing positive [17]. This is a cause of concern as recently, still in Morocco, 95% of non-typhoid *Salmonella* and 80% of typhoid *Salmonella* were transmitted by food [18]. Recently, in South Africa, the prevalence of *Campylobacter* isolates found in children 0–24 months was 13.2% [19]. All these data are alarming and demonstrate the importance of these strains in infectious diseases affecting humans and animals. However, it is clear that in developing countries, particularly those in Central Africa, little data on the involvement and distribution of *Campylobacter* and *Salmonella* or *Shigella* in diarrheal syndromes are available. Thus, we propose in this study to identify the work carried out on the involvement of the genera *Campylobacter*, *Salmonella* and *Shigella* in diarrheal syndromes in Central Africa and to establish a profile of the different prevalence over the period going from 1998 to 2022.

## 2. Methods

This systemic review, carried out using the PRISMA (Preferred Reporting Articles for Systemic Reviews and Meta-Analyses) method, provided an inventory of diarrhoeal infections caused by *Campylobacter*, *Salmonella* and *Shigella* in humans and animals in Central African countries.

### 2.1. Research Environment

According to the United Nations (UN), Central Africa comprises nine (9) countries namely, Angola, Cameroon, Gabon, Equatorial Guinea, the Central African Republic (CAR), the Democratic Republic of Congo (DRC), the Republic of Congo (RC), Sao Tome and Principe, and Chad. Central Africa covers an area of 6,613,000 km^2^, with a population of 163,495,000 inhabitants, or 25 inhabitants per km^2^ in 2017 [20,21]. The region groups multilingual countries with some of the main spoken languages being English, Arabic, French, Portuguese, Lingala, Sango and Spanish.

### 2.2. Literature Search and Identification

The systematic search of the various studies was conducted according to the Preferred Reporting Articles for Systemic Reviews and Meta-Analyses (PRISMA). Perusing five (5) database search engines, namely Pubmed, Google Scholar, Semanticc Scholar, Freefullpdf and Scinapse, we searched for papers published in English, French or Spanish over the period from January 1998 to December 2022, The key words used were “*Campylobacter* diarrheal infection, or *Salmonella* or *Shigella*” or “Prevalence of *Campylobacter* diarrhea, or *Salmonella* or *Shigella*”, or “diarrhea”, or “*Campylobacter*, or *Salmonella* or *Shigella*”, or campylobacteriosis, salmonellosis and shigellosis associated with the names of a country in the study area.

### 2.3. Inclusion and Exclusion Criteria

All original studies of diarrhea with *Campylobacter*, or *Salmonella* or *Shigella* in humans and animals, published between January 1998 and December 2022 were selected. Particular emphasis was placed on the presence of gastrointestinal symptoms and original articless to establish the prevalence of these diarrheal aetiologies. The studies not included were posters, presentations, theses, review and memoirs. All studies without manifestations of gastrointestinal signs, absence of clearly established data on prevalence in the different sub-regions taken into account, and all other diarrheal infections with *Salmonella thyphi* and *Salmonella parathypi* were excluded from this review.

### 2.4. Data Extraction

The data extracted from the articles were then sorted according to the eligibility criteria and the relevance of the data related to the information provided such as the study characteristics (study period, framework, country, study sub-region and design), aetiological agents (*Campylobacter*, *Salmonella*, *Shigella*), study population (subject or domain affected, age of participants, case definition), samples and data analysed (frequencies, percentages, confidence intervals). During this process, all duplicates were eliminated.

### 2.5. Statistical Analysis

The data were extracted and analysed to assess the prevalence of diarrhea with *Campylobacter*, or *Salmonella* or *Shigella* in humans and animals between January 1998 and December 2022 in Central Africa. All data collected during this period was entered in Microsoft office Excel 2010 and sort on Mendeley. A separate analysis was performed for pathogens with the highest frequencies and prevalence.

## 3. Results

### 3.1. Search Results

Searches of the various databases resulted in a total of 400 records (Figure 1). After filtering, 167 articless dealing with diarrhea with *Campylobacter* and/or *Salmonella* or *Shigella* were selected. 150 were ultimately found ineligible and only 17 were deemed eligible. Seventeen articles were selected, including 16 on humans and one on animals. The articless included in this study covered 6 out of 9 countries in Central Africa, namely Gabon (5 articles), Angola (3 articles), Cameroon (3 articles), the Democratic Republic of Congo (3 articles), Chad (2 articles), and the Central African Republic (1 article).

### 3.2. Prevalence of Strains by Country

#### 3.2.1. Mono-Infection

##### *Campylobacter* spp.

Table 1 shows the prevalence of *Campylobacter* spp. in the 9 Central African countries over the years. It was recorded in 4 countries: Angola, Cameroon, DRC and Chad over 4 years in 2008; 2012; 2014 and 2018. These prevalences were recorded over a single year for each of the countries above, Angola in 2018 and DRC in 2014, Cameroon in 2008 and Chad in 2012. The highest prevalence was recorded in DRC (33.3%), followed by Angola (23.0%), Cameroon (9.6%), and the lowest in Chad (3.1%). The other six (6) countries, including Gabon, Equatorial Guinea, CAR, RC, and Sao Tome and Principe, did not have prevalence data of *Campylobacter* involvement in diarrhea over the period from January 1998 to December 2022 considered.

##### *Salmonella* spp.

Table 2 shows the evolution of the prevalence of *Salmonella* spp. in diarrhea by Central African countries and by years. This prevalence was recorded in 5 countries, Angola, Cameroon, Gabon, DRC and Chad over 10 years in 2001; 2008; 2011; 2012, 2015; 2018; 2019; 2020 and 2021. Gabon presented the most data spread over 4 different years, 2001; 2012; 2019 and 2021. It was followed by Cameroon with 3 years’ worth of data (2008, 2018, 2019), Chad with 2 years of data (2011, 20015), and Angola and DRC with only one year of data in 2018 and 2020, respectively. Overall, the highest prevalence was recorded in Gabon (46.6%), followed by the DRC (18.6%), Cameroon (11.2%), and Chad (3.6%). The lowest prevalence of *Salmonella* was found in Angola (2.0%). For the other four (4) countries, namely, Equatorial Guinea, CAR, RC, Sao Tome and Principe, no data of *Salmonella* involvement in diarrhea over the chosen period from January 1998 to December 2022 were found.

##### *Shigella* spp.

Table 3 presents the prevalence of *Shigella* spp. in diarrheal infections by central African country over the years. It is found in 4 countries, Cameroon, Gabon, CAR and Chad over 8 years in 2001; 2008; 2010; 2011; 2012; 2014; 2019 and 2020. Gabon had the most data with 4 datasets spread over 4 different years, 2001, 2012, 2014 and 2020. It is followed by Cameroon with 2 years of data (2008 and 2019), CAR and Chad with only one year of data in 2010 and 2011, respectively. The highest prevalence was recorded in Gabon (44.2%), followed by CAR (9.6%), Cameroon (8.8%), and the lowest in Chad (6.1%). Angola, Equatorial Guinea, DRC, RC, Sao Tome and Principe, had no documented prevalence data of *Shigella* involvement in diarrhea over the chosen period from January 1998 to December 2022.

#### 3.2.2. Co-Infections

##### *Campylobacter-Salmonella* spp. and *Salmonella* or *Shigella* spp.

Table 4 present the prevalence of *Campylobacter*/*Salmonella* spp. and *Salmonella*/*Shigella* spp. co-infections in the 9 Central African countries by year. The data were recorded in only one country, the DRC in 2019. There was a 0.5% prevalence for each co-infection. The other eight (8) including, Angola, Cameroon, Gabon, Equatorial Guinea, CAR, RC, Sao Tome and Principe and Chad, did not present any data recorded over the selected from January 1998 to December 2022.

### 3.3. Comparison of Data Obtained Between Mono and Co-Infection

Table 5 compares the data collected throughout the study for mono- and co-infections. It appears that there were more data collected for mono-infections than for co-infections. In fact, the results revealed that 88.9% of data absent overall were those of co-infections against 44.4% to 55.6% of data for mono-infections. In addition, mono-infections with *Salmonella* spp. presented the most data collected (55.6%) in this study, while, the least recorded data were those of *Campylobacter*/*Salmonella* spp. and *Salmonella*/*Shigella* spp. co-infections with 11.1% each.

### 3.4. Summary of Articles Included in the Journal

Information on the prevalence of *Campylobacter* and *Salmonella* or *Shigella* spp. in diarrheal infections in different Central African countries varied considerably from country to country and from one year to another. The data also varied depending on the bacteria sought after in each study. The country with the most data was Gabon (12 datasets), and Chad (4 datasets) was the one with the least data. Out of 17 articless, 3 (17.6%) articless used as diagnostic tools PCR or Real time PCR versus 14 (82.4%) articless which used conventional diagnosis (culture) (Table 6).

## 4. Discussion

This study reviewed the work carried out on the involvement of the genera *Campylobacter*, *Salmonella* and *Shigella* spp. in diarrheal syndromes in Central Africa in order to establish a profile of the different prevalence over the period 1998 to 2022.

The seventeen works recorded in this study came from 6 of the 9 countries of Central Africa, namely Gabon (5 articles), Angola (3 articles), Cameroon (3 articles), the Democratic Republic of Congo (3 articles), Chad (2 articles) and Central African Republic (1 item). The results obtained are similar to those of Oppong et al.(2020). In fact, this study reported a clear lack of data on infectious pathogens associated with diarrhea in the sub-Saharan Africa region, particularly in the Central Africa region. One likely cause would be that this sub-region is not only the second in Africa with the fewest countries, but also has a low capacity to diagnose germs [39,40]. This argument is corroborated, in this study, by the absence or lack of records over a period of two from January 1998 to December 2022, depending on the type of infection. Moreover, several studies have shown that the variables in the data collection depend in particular on the causative agents, the type of disease or infection, the criteria for inclusion and exclusion of the study, the subject (human, animal or plant), and the region of the world in which the study was conducted [40,41].

Also, co-infections accounted for 88.9% of the missing data, compared to 44.4% to 55.6% for mono-infections. These results could be explained by the fact that conventional culture methods are commonly used for routine diagnosis in developing countries [41,42,43]. However, several studies would establish a poly-microbial profile and or associated with viruses in the panel of agents responsible for gastroenteritis in the world [44,45] which would show the significant proportion of co-gastroenteritis infections. Moreover, they would be largely underestimated in research by conventional methods [46,47].The impact of this finding on public health strategy would be, firstly, a bias in the diagnosis, monitoring and therapy of diarrheal infections. Numerous studies have already pointed to a poly-microbial, bacterial, parasitic and viral mix in diarrheal infections [48]. Consequently, the introduction of more robust and effective diagnostic tools for the simultaneous detection of these pathogens via molecular biology would be judicious, and would enable therapeutic regimens to be modified and adapted in the event of co-infections [3,17,18].

In addition, the fact that the most frequent data (55.6%) concern *Salmonella* spp. mono-infections is in line with research by Chow et al. (2010), which shows that the genera *Salmonella, Campylobacter* and *Shigella* are among the main causes of gastroenteritis in developing and low-income countries [49].

Although discontinuous, this trend appears to be relatively corroborated in this study. Indeed, diarrheal infections with *Salmonella* and *Shigella* spp. seem to have a fairly continuous follow-up over time and in different Central African countries. It is discontinuous over 10 years in 5 countries, Angola, Cameroon, Gabon, Democratic Republic of Congo (DRC) and Chad for mono infections with *Salmonella* spp., over 8 years in 4 countries, Cameroon, Gabon, Central African Republic (CAR) and Chad for *Shigella* spp. These results could be explained firstly, by the fact that non-typhoid *Salmonella* are among the most common food and water zoonoses affecting humans in the world [1] and their diagnosis by conventional methods of culture in liquid and solid media is not tedious [48]. Also, the widening and ease of use of diagnostic tools of the genus *Salmonella* usually in double detection with the genus *Shigella*, facilitate and explain widely its place in the panel of detection of pathogens causing diarrhea in animal and human world [50,51].

Nevertheless, these results disagree with studies conducted in Europe. In fact, thanks to a quasi-permanent surveillance, with advanced diagnostic tools, and almost 135 laboratories involved in the surveillance and monitoring of *Salmonella* infections in France alone, it is established that in France and the rest of Europe, *Salmonella* would be considered the second cause of food zoonosis in 67.2% of cases in humans, behind the *Campylobacter* genus [52,53].

In the same order, in 2022, the World Health Organization reported an epidemic of *Salmonella enterica* serovar Typhimurium caused by the consumption of chocolate products from Belgium, affecting more than 11 European countries including the United Kingdom with 65 cases of salmonellosis [53]. The lack of surveillance protocol and the low number of laboratories involved in monitoring in the Central African sub-region would cause a rather discontinuous and incomplete record of data as observed in this study [54].

Moreover, diarrheal *Campylobacter* infection and the resulting co-infections have the fewest data. Mono-infection with *Campylobacter* was recorded in 4 countries, Angola, Cameroon, DRC, and Chad over 3 years compared to a single study in DRC in 2019 reporting co-infections with *Campylobacter*/*Salmonella* spp. These results could be justified by the fact that *Campylobacter* are slow-growing organisms, tedious to identify, often requiring very selective and expensive media (sometimes unavailable in the sub-region) and sufficiently equipped and experienced laboratories for their diagnostics [55].

In addition, most of the studies identified on the continent are most often conducted in Arab countries and West African countries, particularly in the field of livestock [52,53] because of international aid programmes in agriculture and livestock, via the Food and Agriculture Organization (FAO) of the United Nations, for instance [56].

Furthermore, according to the literature, the contamination or spread *Campylobacter* infections in humans is due to the excrement of farm animals close to households, farms and slaughterhouses, particularly affecting children and vulnerable people, in developing countries [57,58]. By contrast, in developed countries, it is due to the consumption of milk, meat and unpasteurized poultry products [41,59]. Climate change may also facilitate the spread of *Campylobacter* spp. infection around the world. According to Kuhn et al. (2020), the rate of *Campylobacter* spp. infections in four Northern European countries; Denmark, Finland, Norway and Sweden, could increase by 25% by the end of the 2040s and by 196% by the end of the 2080s compared to the 2000–2015 baseline [60].

As for the origins of the infections, in some parts of Asia, notably in Japan, 68% to 75% of cases of food poisoning in *Campylobacter* are reported to have occurred in restaurants, most of which originated from sushi filled with raw or undercooked chicken meat [61].

All these studies show the complexity surrounding *Campylobacter* infections and highlight the fact that research into these agents depends on the area, the country and the research means available. The glaring absence of data on *Campylobacter* infections in Central Africa requires that more attention is paid to this sub-region of Africa.

This work also showed that the conventional culture method is the most predominant type of diagnosis (82.35%) compared to real-time PCR or PCR (17.65%). This strong predominance of the conventional culture method is due to the fact that it is the oldest method of diagnosis of enteric bacterial pathogens and remains the reference method for the isolation of bacterial strains necessary for antimicrobial testing. These results, as already mentioned, explain the low prevalence rate of recorded *Campylobacter*. Moreover, numerous studies have shown that, the conventional culture method has a low sensitivity (37.2%) of detection of *Campylobacter* and other pathogens compared to PCR (96.7%) [62]. Although most research report stool detection for *Campylobacter* and *Salmonella* genera, research using molecular biology methods remain more sensitive and increase the detection rate of these germs [41].

## 5. Conclusions

The impact of diarrhea caused by *Campylobacter*, *Salmonella* and *Shigella* in Central Africa cannot be easily assessed, as this study highlights a notable lack of data on the involvement of these bacteria in human and animal diarrhea in Central Africa. The prevalence data obtained varied according to the causative agent of diarrhea and presented a discontinuous spatio-temporal distribution. On the other hand, studies on the *Campylobacter* genus and associated infections are almost non-existent throughout this investigation, perhaps because the culture technique on agar medium remains the technique widely used in diagnosis in these countries due to the high cost related to this research.

In view of our conclusions highlighting a serious lack of data on diarrheal infections caused by *Campylobacter* and *Salmonella* or *Shigella* in humans and animals in Central Africa in the period studied. it is urgent to set up a database for the sub-region on diarrheal infections caused by *Campylobacter* and *Salmonella* or *Shigella*.

Despite the paucity of recorded data on the prevalence of diarrheal infections due to *Campylobacter* in this sub-region, it is crucial that scientific studies focus on the diagnosis and monitoring of this zoonotic bacterium. Also, improved diagnosis will necessarily involve the integration of molecular tools in the diagnosis of these diarrheic syndromes in both humans and animals.

In addition, Central African countries need more funding, qualified people and state-of-the-art, well-equipped laboratories for research into the causative agents of gastroenteritis, and in particular for research into *Campylobacter*, which is still relatively unknown compared to other enteric pathogens.

## Figures and Tables

**Figure 1 ijerph-21-01635-f001:**
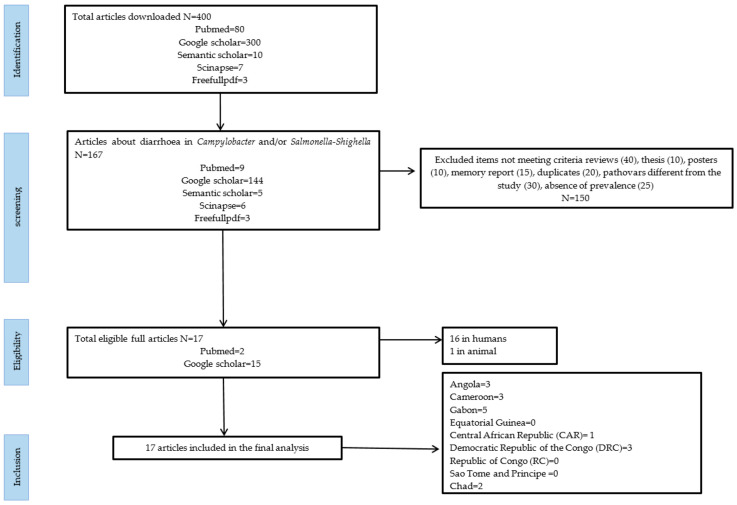
PRISMA Study Selection Flowchart.

**Table 1 ijerph-21-01635-t001:** Summary of the prevalence of *Campylobacter* spp. in diarrheal infections in different Central African countries over the selected years.

	Countries/Year	1998 to 2007	2008	2009 to 2011	2012	2013	2014	2015 to 2017	2018	2019 to 2022
***Campylobacter*** **spp.**	**Angola**	0	0	0	0	0	0	0	**23.0%**	0
	**Cameroon**	0	**9.6%**	0	0	0	0	0	0	0
	**Gabon**	0	0	0	0	0	0	0	0	0
	**Equatorial Guinea**	0	0	0	0	0	0	0	0	0
	**CAR**	0	0	0	0	0	0	0	0	0
	**DRC**	0	0	0	0	0	**33.3%**	0	0	0
	**RC**	0	0	0	0	0	0	0	0	0
	**Sao Tome and Principe**	0	0	0	0	0	0	0	0	0
	**Chad**	0	0	0	**3.1%**	0	0	0	0	0

**0**: of data available to establish prevalence; **%**: prevalence by year and country; Central African Republic (CAR); Democratic Republic of Congo (DRC); Republic of Congo (RC).

**Table 2 ijerph-21-01635-t002:** Summary of the prevalence of *Salmonella* spp. in diarrheal infections in different Central African countries over the selected years.

	Countries/Year	1998 to 2000	2001	2002 to 2007	2008	2009 to 2010	2011	2012	2013 to 2014	2015	2016 to 2017	2018	2019	2020	2021	2022
***Salmonella*** **spp.**	**Angola**	0	0	0	0	0	0	0	0	0	0	**2.0%**	0	0	0	0
	**Cameroon**	0	0	0	**11.2%**	0	0	0	0	0	0	**16.0%**	**9.8%**	0	0	0
	**Gabon**	0	**46.6%**	0	0	0	0	**40.8%**	0	0	0	0	0	**11.6%**	**2.1%**	0
	**Equatorial Guinea**	0	0	0	0	0	0	0	0	0	0	0	0	0	0	0
	**CAR**	0	0	0	0	0	0	0	0	0	0	0	0	0	0	0
	**DRC**	0	0	0	0	0	0	0	0	0	0	0	0	**18.6%**	0	0
	**RC**	0	0	0	0	0	0	0	0	0	0	0	0	0	0	0
	**Sao Tome and Principe**	0	0	0	0	0	0	0	0	0	0	0	0	0	0	0
	**Chad**	0	0	0	0	0	**3.1%**	0	0	**9.6%**	0	0	0	0	0	0

**0**: of data available to establish prevalence; **%**: prevalence by year and country; Central African Republic (CAR); Democratic Republic of Congo (DRC); Republic of Congo (RC).

**Table 3 ijerph-21-01635-t003:** Summary of the prevalence of *Shigella* spp. in diarrheal infections in different Central African countries over the selected years.

	Countries/Year	1998 to 2000	2001	2002 to 2007	2008	2009	2010	2011	2012	2013	2014	2015 to 2018	2019	2020	2021 to 2022
***Shigella*** **spp.**	**Angola**	0	0	0	0	0	0	0	0	0	0	0	0	0	0
	**Cameroon**	0	0	0	**8.2%**	0	0	0	0	0	0	0	**2.0%**	0	0
	**Gabon**	0	**44.2%**	0	0	0	0	0	**27.8%**	0	**33.3%**	0	0	**3.6%**	0
	**Equatorial Guinea**	0	0	0	0	0	0	0	0	0	0	0	0	0	0
	**CAR**	0	0	0	0	0	**9.6%**	0	0	0	0	0	0	0	0
	**DRC**	0	0	0	0	0	0	0	0	0	0	0	0	0	0
	**RC**	0	0	0	0	0	0	0	0	0	0	0	0	0	0
	**Sao Tome and Principe**	0	0	0	0	0	0	0	0	0	0	0	0	0	0
	**Chad**	0	0	0	0	0	0	**6.1%**	0	0	0	0	0	0	0

**0**: of data available to establish prevalence; **%**: prevalence by year and country; Central African Republic (CAR); Democratic Republic of Congo (DRC); Republic of Congo (RC).

**Table 4 ijerph-21-01635-t004:** Summary of the *Campylobacter/Salmonella* spp. and *Salmonella/Shigella* spp. in the different countries of Central Africa over the selected years.

	Countries/Year	1998 to 2018	2019	2020 to 2022
** *Campylobacter* ** **/*Salmonella* spp.** ** *Salmonella* ** **/*Shigella* spp.**	**Angola**	0	0	0
	**Cameroon**	0	0	0
	**Gabon**	0	0	0
	**Equatorial Guinea**	0	0	0
	**CAR**	0	0	0
	**DRC**	0	**0.5%**	0
	**RC**	0	0	0
	**Sao Tome and Principe**	0	0	0
	**Chad**	0	0	0

**0**: of data available to establish prevalence; **%**: prevalence by year and country; Central African Republic (CAR); Democratic Republic of Congo (DRC); Republic of Congo (RC).

**Table 5 ijerph-21-01635-t005:** Comparison of the prevalence rate obtained between mono and co-infection.

		Numbers of Countries, n (N_t_ = 9 Countries)	Prevalence (%) (n/N_t_) × 100	Absence of Prevalence (%) (n_ap_/N_t_) × 100	Total
Mono-Infection	*Campylobacter* spp.	4	44.4	55.6	100.0
	*Salmonella* spp.	5	55.6	44.4	100.0
	*Shigella* spp.	4	44.4	55.6	100.0
Co-infection	*Campylobacter/Salmonella* spp.	1	11.1	88.9	100.0
	*Salmonella/Shigella* spp.	1	11.1	88.9	100.0

**N_t_**: total numbers of countries in Central Africa, **n**: total number of countries with prevalence by type of infection, **n_ap_**: total number of countries with no prevalence by type of infection.

**Table 6 ijerph-21-01635-t006:** Summary of prevalence of *Campylobacter* and *Salmonella*-*Shigella* spp. in countries of the Central African region; presence of pathogen, disease, symptoms, year of work, subject, age; sample, prevalence, provenance, diagnosis, journal and reference.

Countries	Pathogens	Affection	Symptoms	Year	Patients	Age	Samples	Prevalence (%)	Origin	Diagnostic	Newspaper Source	Refs.
**Angola**	*Campylobacter* spp.	diarrhea	malnutrition/dehydration	2014	Child	under 5 years	Feces	23	Hospital	PCR	Pubmed	[22]
	*Salmonella* spp.	diarrhea	malnutrition/dehydration	2014	Child	under 5 years	Feces	2	Hospital	PCR	Pubmed	
	*Campylobacter* spp.	diarrhea	malnutrition/dehydration	2014	Child	5 years	Feces	0	Hospital	Conventional cultivation	Google scholar	[23]
	*Salmonella* spp.	diarrhea	malnutrition/dehydration	2014	Child	5 years	Feces	0	Hospital	Conventional cultivation	Google scholar	
	*Shigella* spp.	diarrhea	malnutrition/dehydration	2014	Child	5 years	Feces	0	Hospital	Conventional cultivation	Google scholar	
	*Campylobacter* spp.	diarrhea	malnutrition/dehydration	2016	Child	under 5 years	Blood	0	Hospital	PCR	Google scholar	[24]
	*Salmonella* spp.	diarrhea	malnutrition/dehydration	2016	Child	under 5 years	Blood	0	Hospital	PCR	Google scholar	
	*Shigella* spp.	diarrhea	malnutrition/dehydration	2016	Child	under 5 years	Blood	0	Hospital	PCR	Google scholar	
**Cameroon**	*Campylobacter jeujeni*	diarrhea	asymptomatic	2008	Child	under 5 years	Feces	9.6	Hospital	Conventional cultivation	Google scholar	[25]
	*Salmonella* spp.	diarrhea	asymptomatic	2008	Child	under 5 years	Feces	11.2	Hospital	Conventional cultivation	Google scholar	
	*Shigella* spp.	diarrhea	asymptomatic	2008	Child	under 5 years	Feces	8.8	Hospital	Conventional cultivation	Google scholar	
	*Salmonella* spp.	diarrhea	asymptomatic	2018	Child	0 at 5 years	Feces	16	Hospital	Conventional cultivation	Google scholar	[26]
	*Shigella* spp.	diarrhea	asymptomatic	2019	Child	1 at 5 years	Feces	2	Hospital	Conventional cultivation	Google scholar	
	*Salmonella enterica*	diarrhea	stomach upset/fever/vomiting	2019	Adults/children	0 at 60 years	Feces	9.8	Hospital	Conventional cultivation	Google scholar	[27]
**Gabon**	*Salmonella enterica*	diarrhea	stomach upset/fever/vomiting	2020	Child	under 5 years	Feces	3	Hospital	Conventional cultivation	Google scholar	[28]
	*Salmonella* spp.	diarrhea	stomach upset/fever/vomiting	2020	Child	under 5 years	Feces	8.6	Hospital	Conventional cultivation	Google scholar	
	*Shigella* spp.	diarrhea	stomach upset/fever/vomiting	2020	Child	under 5 years	Feces	1.5	Hospital	Conventional cultivation	Google scholar	
	*Shigella sonei*	diarrhea	stomach upset/fever/vomiting	2020	Child	under 5 years	Feces	1	Hospital	Conventional cultivation	Google scholar	
	*Salmonella* spp.	acute diarrhea	fever/digestive signs	2001	Adults	18 years to older	Feces	46.6	Hospital	Conventional cultivation	Google scholar	[29]
	*Shigella* spp.	acute diarrhea	fever/digestive signs	2001	Adults	19 years to older	Feces	44.2	Hospital	Conventional cultivation	Google scholar	
	*Salmonella* spp.	acute diarrhea	fever/digestive signs	2012	Child	0 at 15 years	Feces	40.8	Hospital	Conventional cultivation	Google scholar	[30]
	*Shigella* spp.	acute diarrhea	fever/digestive signs	2012	Child	0 at 15 years	Feces	27.8	Hospital	Conventional cultivation	Google scholar	
	*Salmonella* spp.	acute diarrhea	total symptomatic	2021	Child	under 5 years	Feces	2.1	Hospital	Real-time PCR	Google scholar	[31]
	*Shigella flexeri*	acute diarrhea	blood in stool/cramps	2014	Child	1 at 2 years	Feces	78	Hospital	Conventional cultivation	Google scholar	[32]
	*Shigella boydii*	acute diarrhea	blood in stool/cramps	2014	Child	1 at 2 years	Feces	14	Hospital	Conventional cultivation	Google scholar	
	*Shigella sonnei*	acute diarrhea	blood in stool/cramps	2014	Child	1 at 2 years	Feces	8	Hospital	Conventional cultivation	Google scholar	
**CAR**	*shigella* spp.	acute diarrhea	sang dans les selle/mucus/liquide	2010	Child	5 years	Feces	9,6	Hospital	Conventional cultivation	Google scholar	[33]
**DRC**	*Salmonella/shigella* spp.	acute diarrhea	malnutrition/dehydration	2019	Child t	77 month	Feces	0.54	Hospital	Conventional cultivation	Google scholar	[34]
	*Campylobacter/Salmonella* spp.	acute diarrhea	malnutrition/dehydration	2019	Child t	77 month	Feces	0.54	Hospital	Conventional cultivation	Google scholar	
	*Campylobacter* spp.	diarrhea	asymptomatic	2014	Goat		Feces	33.3	Farm	Conventional cultivation	Google schola	[35]
	*Salmonella non typhique*	diarrhea	unspecified	2020	Child	under 5 years	Blood	2.1	Hospital	Conventional cultivation	Google scholar	[36]
	*Salmonella non typhique*	diarrhea	unspecified	2020	Child	under 5 years	Feces	35.1	Hospital	Conventional cultivation	Google scholar	
**Chad**	*Salmonella non typhique*	diarrhea	unspecified	2015	Adults/children	not specified	Feces	9.6	Hospital	Conventional cultivation	Google scholar	[37]
	*Campylobacter* spp.	diarrhea	Vomiting/asthenia/fever	2011	Adults	25 years to older	Feces	3.1	Military Camp	Conventional cultivation	Google scholar	[38]
	*Salmonella* spp.	diarrhea	Vomiting/asthenia/fever	2011	Adults	25 years to older	Feces	3.1	Military Camp	Conventional cultivation	Google scholar	
	*Shigella* spp.	diarrhea	Vomiting/asthenia/fever	2011	Adults	25 years to older	Feces	6.1	Military Camp	Conventional cultivation	Google scholar	

## Data Availability

The datasets generated and/or analysed during the current study are available in the SearchRix repository, https://www.cabidigitallibrary.org/doi/10.1079/searchRxiv.2023.00258, accessed on 3 July 2023.

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
