# Peer review of "The Impact of Campylobacter, Salmonella, and Shigella in Diarrheal Infections in Central Africa (1998–2022): A Systematic Review"

_ijerph, 2024, doi:10.3390/ijerph21121635_

Round 1

Reviewer 1 Report

Comments and Suggestions for Authors

Comments on the Quality of English Language

Author Response

1- Clarify in the title that this manuscript is a systematic review.

Answer: Correct in the manuscript.

2-: What was your specific data base, need to confirm!

Answer: Our specific database was Pubmed. However, searches were also carried out on several other databases such as Google Scholar, Semantic Scholar, Freefullpdf, and Scinapse, in order not only to broaden searches but also to compensate for the fact that many African publications do not necessarily appear in the PubMed database. The datasets generated and/or analysed during the current study are available in the SearchRix repository,https://www.cabidigitallibrary.org/doi/10.1079/searchRxiv.2023.00258 (line 388-390)

3-What was your whole search strategy and key boards? Mention in the method

Answer: Correct in the manuscript.

4- Perform the quality assessment of your included paper

Answer: To fit in with our research objectives, we placed particular emphasis on original articles mentioning gastrointestinal symptoms and presenting clearly established prevalences of Campylobacter, Salmonella and Shigella diarrhoeal infections in humans and animals in Central African countries. This type of research is the most widely used means of promoting and consolidating scientific research, but it is also an indicator of performance and the originality of ideas and knowledge techniques on many subjects that can undermine a region. Moreover, the scientific quality of other types of manuscripts, theses, posters and dissertations often takes second place to original articles.

Reviewer 2 Report

Comments and Suggestions for Authors

Dear Authors,

Your Systematic Review entitled "Involvement and impact of Campylobacter and Salmonella-Shigella in diarrheal infections in Central Africa from 1998 to 2022" has been reviewed,

In fact, this review highlights an important topic related to the prevalence and the impact pf three main bacterial causing-diarrhea agents in a developing region of the world mainly "in the Central Africa region", during a long period of time, a 25 years, between 1998 and 2022.

The paper is well written in English, with a good design, but unfortunately I have several comments regarding it.

Kindly find below the list of my Minor and Major Comments regarding this work:

Minor Comments:

01- In the whole manuscript authors are invited to put bacterial names in italic (Genera and Species).

02- In the "Abstract" line 19, authors are invited to replace the term "studies" by another term, to minimize duplication.

03- In the "Abstract" line 20, authors are invited to put space after the point before "Methods".

04- In the "Abstract" line 25, authors are invited to replace 5 by 6, since there is info from 6 countries and not from 5.

05- In the whole manuscript, when authors talked about Campylobacter, Salmonella and Shigella, they are invited to separate Salmonella from Shigella. (Try to use Salmonella or Shigella, instead of Salmonella-Shigella). 

06- In the "Introduction" line 41, Concerning the reference "03", This is an old reference date from 2015, authors are invited to put a recent one.

07- In the "Introduction" line 49, Concerning the reference "06", The reference is not suitable they can use the following one:

-- https://pubmed.ncbi.nlm.nih.gov/34316469/

08- In the "Introduction" line 50, Concerning the reference "07", The reference is not suitable, I suggest to replace it by:

-- https://www.ncbi.nlm.nih.gov/pmc/articles/PMC7119329/

09- In the "Introduction" lines 50-53, The idea mentioned in this senetence is not specific to the topic, E. coli is nopt considered as the main enteropathogenic bacteria, I suggest to reformulate the sentence.

In Addition, the reference is not suitable at all for the idea, therefore i suggest the following one:

-- https://www.ncbi.nlm.nih.gov/pmc/articles/PMC8793005/

10- In the whole manuscript, When talking about the virulence of Salmonella virulence it is good to use the following reference:

-- https://www.mdpi.com/2036-7481/14/3/99

11- In the "Introduction" lines 56-57, authors are kindly invited to add more references for this idea, I suggest the following one:

-- https://pubmed.ncbi.nlm.nih.gov/32032018/

12- In the Whole manuscript when using "Campylobacter, or Salmonella-Shigella" I suggest to replace it by "Campylobacter, Salmonella, or Shigella"

13- In the "Results" line 146, at the end of the sentence it is "1 article" instead of "1 articles".

14- For the following Tables (1-5), Tables should be below the text and not before the text.

15- In the Table 5, authors are invited to replace "pays" by "countries".

16- In the "Discussion" line 241, authors are invited To add the following references in addition to 42:

-- https://www.ncbi.nlm.nih.gov/pmc/articles/PMC10082710/

-- https://journals.asm.org/doi/full/10.1128/cmr.00006-15

17- In the "Discussion" line 243, authors are invited To add the following references in addition to 43:

-- https://synapse.koreamed.org/articles/1098074

18- In the "Discussion" line 245, authors are invited To add the following references in addition to 44:

-- https://link.springer.com/article/10.1186/s12887-019-1513-8

19- In the "Discussion" line 246, authors are invited to remove the letter "d".

20- In the "Discussion" line 261, authors are invited To add the following references in addition to 46:

-- https://www.ncbi.nlm.nih.gov/pmc/articles/PMC10750756/

Major Comments:

01- In the "Materials" Lines 107-115, When searching for articles I think that several keywords most be used such as campylobacteriosis, salmonellosis and shigellosis.

02- Concerning the whole study, do you think that with this low number of studies in this systematic review, the impact of this paper will be of a high value?

03- Concerning the solo paper concerning animals, I suggest to remove it from this review.

BR,

Author Response

Minor Comments:

1-In the whole manuscript authors are invited to put bacterial names in italic (Genera and Species).

Answer: Correct in the manuscript.

2- In the "Abstract" line 19, authors are invited to replace the term "studies" by another term, to minimize duplication

Answer: Correct in the manuscript.

3-: In the "Abstract" line 20, authors are invited to put space after the point before "Methods".

Answer: Correct in the manuscript.

4- In the "Abstract" line 25, authors are invited to replace 5 by 6, since there is info from 6 countries and not from 5.

Answer: Correct in the manuscript.

5-In the whole manuscript, when authors talked about Campylobacter, Salmonella and Shigella, they are invited to separate Salmonella from Shigella.(Try to use Salmonella or Shigella, instead of Salmonella-Shigella).

Answer: corrected in the manuscript, except for the notion of co-infection represented by for Salmonella/Shigella or Campylobacter/Salmonella.

6-In the "Introduction" line 41, Concerning the reference "03", This is an old reference date from 2015, authors are invited to put a recent one.

Answer: Correct in the manuscript.

7-In the "Introduction" line 49, Concerning the reference "06", The reference is not suitable they can use the following one: - https://pubmed.ncbi.nlm.nih.gov/34316469/

Answer: Correct in the manuscript.

8-In the "Introduction" line 50, Concerning the reference "07", The reference is not suitable, I suggest to replace it by:- https://www.ncbi.nlm.nih.gov/pmc/articles/PMC7119329/.

Answer: Correct in the manuscript.

9- In the "Introduction" lines 50-53, The idea mentioned in this senetence is not specific to the topic, E. coli is not considered as the main enteropathogenic bacteria. I suggest to reformulate the sentence. In Addition, the reference is not suitable at all for the idea, therefore I suggest the following one:- https://www.ncbi.nlm.nih.gov/pmc/articles/PMC8793005/

Answer: Indeed, despite the fact that bacteria account for 50-60% of microorganisms isolated during acute diarrheal episodes, most studies in Black Africa seem to focus primarily on diarrheal infections caused by Escherichia coli Diarrheic, to the detriment of other genera such as Campylobacter, Salmonella spp, and Shigella spp.

Correct in the manuscript.

10-In the whole manuscript, When talking about the virulence of Salmonella virulence it is good to use the following reference:- https://www.mdpi.com/2036-7481/14/3/99

Answer: Correct in the manuscript. However, our study did not focus on the virulence or virulence factors of Salmonella.

11-In the "Introduction" lines 56-57, authors are kindly invited to add more references for this idea, I suggest the following one:- https://pubmed.ncbi.nlm.nih.gov/32032018/

Answer: Correct in the manuscript.

12-In the Whole manuscript when using "Campylobacter, or Salmonella-Shigella" I suggest to replace it by "Campylobacter, Salmonella, or Shigella".

Answer: Correct in the manuscript.

13-In the "Results" line 146, at the end of the sentence it is "1 article" instead of "1 articles".

Answer: Correct in the manuscript.

14-For the following Tables (1-5), Tables should be below the text and not before the text.

Answer: Correct in the manuscript.

15-In the Table 5, authors are invited to replace "pays" by "countries".

Answer: Correct in the manuscript.

16- In the "Discussion" line 241, authors are invited To add the following references in addition to 42:- https://www.ncbi.nlm.nih.gov/pmc/articles/PMC10082710/

-- https://journals.asm.org/doi/full/10.1128/cmr.00006-15

Answer: Correct in the manuscript.

17- In the "Discussion" line 243, authors are invited To add the following references in addition to 43:- https://synapse.koreamed.org/articles/1098074.

Answer: Correct in the manuscript.

18- In the "Discussion" line 245, authors are invited To add the following references in addition to 44:- https://link.springer.com/article/10.1186/s12887-019-1513-8.

Answer: Correct in the manuscript.

19-A : - In the "Discussion" line 246, authors are invited to remove the letter "d".

Answer: Correct in the manuscript.

20-In the "Discussion" line 261, authors are invited To add the following references in addition to 46:- https://www.ncbi.nlm.nih.gov/pmc/articles/PMC10750756/

Answer: Correct in the manuscript.

Major Comments:

1- In the "Materials" Lines 107-115, When searching for articles I think that several keywords most be used such as campylobacteriosis, salmonellosis and shigellosis.

Answer: These words were also used for research, but were not productive enough: 30 articles were registered compared with 370 articles for the rest of the keywords. We have included the corrections in the manuscript.

2-Concerning the whole study, do you think that with this low number of studies in this systematic review, the impact of this paper will be of a high value?

Answer: The small number of studies or articles included in our research does not limit its quality. On the contrary, the small number of studies and articles in our database highlights the need for scientific research into diarrhoeal infections caused by these bacterial agents, particularly Campylobacter, in the Central African sub-region. These results provide information on the areas of study being tackled by various research and health centers. In addition, this low level is a cause for concern in the Central African sub-region, given the knowledge and surveillance measures in place. However, elsewhere in Europe and in industrialized countries, surveillance programs, control measures and diagnostic methods are highly developed for these different pathogens, in terms of human health, livestock farming and the environment.

3-Concerning the solo paper concerning animals, I suggest to remove it from this review.

Answer: The withdrawal of the only study on animals does not seem to us to be judicious, because in view of the importance of livestock farming and the consumption of imported meat in the countries of this region, it is rather worrying to find only one study on these agents. These various bacterial agents, Campylobacter, Salmonella and Shigella, are normally among the leading causes of abortion, death and contamination in animals, and are also vectors of transmission to humans during food and water-borne outbreaks worldwide. So how is it that only one study is listed in our research should be the question.

Reviewer 3 Report

Comments and Suggestions for Authors

The manuscript entitled "Involvement and Impact of Campylobacter and Salmonella-Shigella in Diarrheal Infections in Central Africa from 1998 to 2022" focuses on the prevalence of Campylobacter, Salmonella, and Shigella in diarrheal infections in Central Africa from 1998 to 2022. These infections are a significant public health issue, particularly in developing countries. The authors conducted a systematic review using five databases to identify relevant studies. The review included studies from nine Central African countries over 25 years. Seventeen articles were selected, with data from six countries. Salmonella was the most common mono-infection, followed by Campylobacter and Shigella. Co-infections were less frequently reported. With some necessary improvements, the manuscript can reach its full potential and be ready for publication.

Following are the comments.

Line 13 -137: Which variables were used for statistical analysis and what was tested? Specify.

In relation to the findings, figure 1 illustrates that the meta-analysis encompassed 17 studies from six of the nine Central African countries: Gabon, Angola, Cameroon, the Democratic Republic of Congo, Chad, and the Central African Republic. However, the results section references all nine countries, indicating zero outcomes for the remaining three. It is essential to clarify whether any studies yielded zero results for these countries. If not, the focus should be restricted to the countries where studies were performed, and it is recommended that the revised manuscript reflect any changes in statistical methodology that may arise from this focus.

Table 6: It is recommended to incorporate a horizontal border separating each country within Table 6.

Line 212: The authors mentioned that Gabon emerged as the country with the highest volume of data, comprising a total of 12 datasets. Additionally, the study features five articles carried out in Gabon, necessitating clarification on this aspect.

Revise line 246; it needs a grammatical check.

Comments on the Quality of English Language

Minor English editing is required.

Author Response

1-Which variables were used for statistical analysis and what was tested? Specify.

Answer: We have not performed any statistical tests in this manuscript, as the aim was to compile pre-existing data on diarrhoeal infections with Campylobacter, Salmonella and Shigella in the countries of the Central African region between 1998 and 2022. For this purpose, the data were unprocessed, apart from a presentation of raw data by year and by country.

2-In relation to the findings, figure 1 illustrates that the meta-analysis encompassed 17 studies from six of the nine Central African countries: Gabon, Angola, Cameroon, the Democratic Republic of Congo, Chad, and the Central African Republic. However, the results section references all nine countries, indicating zero outcomes for the remaining three. It is essential to clarify whether any studies yielded zero results for these countries. If not, the focus should be restricted to the countries where studies were performed, and it is recommended that the revised manuscript reflect any changes in statistical methodology that may arise from this focus.

Answer: The manuscript reports on the involvement and impact of the agents Campylobacter, Salmonella and Shigella in diarrhoeal infections in the 9 countries of the Central African sub-region between 1998 and 2022. However, the results in Figure 1 show that only six (6) of the nine countries presented usable data in accordance with the inclusion and exclusion criteria of our study (line 132-140). The other three countries (CAR, RC and Sao Tome and Principe) did not have compliant data and clearly established prevalences, hence the null results indicated by zeros in the various tables. In the end, the studies gave zero results for these countries (CAR, RC and Sao Tome and Principe).

3-Table 6: It is recommended to incorporate a horizontal border separating each country within Table 6.

Answer: Correct in the manuscript.

4-Line 212: The authors mentioned that Gabon emerged as the country with the highest volume of data, comprising a total of 12 datasets. Additionally, the study features five articles carried out in Gabon, necessitating clarification on this aspect.

Answer: Indeed, 5 articles have been listed in Gabon, with references ranging from 27 to 31. However, the aim of our study is to establish the prevalence of three bacterial agents: Campylobacter, Salmonella and Shigella, we particularly looked by studies (article) if these prevalences are presented. As some authors sometimes deal with several causal agents in their research, we end up with articles that provide data for all three agents at the same time, or their variants. As a result, for reference 27 (4 data), 28 (2 data), 29 (2 data), 30 (1 data) and 31 (3 data), there is a total of 12 data for 5 articles.

5-Revise line 246; it needs a grammatical check.

Answer: These results could be explained by the fact that conventional culture methods are the most widely used for routine diagnosis and research in developing countries, and are less sensitive than molecular methods.

Correct in the manuscript.

Reviewer 4 Report

Comments and Suggestions for Authors

Review report  for : Involvement and Impact of Campylobacter and Salmonella-Shi- 2 gella in Diarrheal Infections in Central Africa from 1998 to 2022

This study reviewed the prevalence of Campylobacter, Salmonella, and Shigella spp. in diarrheal infections across Central Africa from 1998 to 2022. The seventeen studies identified were from six of the nine Central African countries, highlighting a significant data gap in this region. Authors have clearly stated the background and aim of the study,  and provided a concise overview of the methods used to search for relevant studies.  

 My main concern is that the authors' aim of the study is to identify the research conducted on the involvement of the genera Campylobacter, Salmonella, and Shigella in diarrheal syndromes in Central Africa and to establish a profile of their prevalence over the period from 1998 to 2022. However, the reasons for the limited number of such studies may be related to a lack of records on human and animal cases of infections caused by these bacteria. The authors have analyzed the work of 17 studies, but they need to explain why there are so few studies, considering factors such as fewer cases and potentially higher immune responses in the study populations compared to European populations. Although they mention European cases, comparing these with South African countries is challenging due to differences in individual responses to infections. This issue needs to be addressed. Therefore, I suggest the following improvements and recommend considering this as a mini review.

Areas to be improved

Title: My suggestion for the title is “Impact of Campylobacter and Salmonella-Shigella in Diarrheal Infections in Central Africa (1998-2022).

Abstract: The abstract lacks specific details about the analysis performed on the selected studies. Consider adding a sentence or two to describe the inclusion and exclusion criteria used to select the studies. Provide more detail about the diagnostic tools used in the selected studies. For example, what types of PCR tests were used, and what were the sensitivities and specificities of these tests? The presentation of the data could be improved. The conclusion is quite brief and doesn't provide much insight into the implications of the study's findings. Consider expanding on the conclusion to discuss the significance of the study's results and potential future directions.  Consider including a statement about the limitations of the study, such as the potential for bias in the selected studies or the lack of data on certain aspects of the topic.

Introduction: Overall, the introduction provides a good overview of the importance of food-borne and water-borne diseases, highlighting the significant morbidity and mortality they cause worldwide. The text also effectively sets the stage for the systematic review by emphasizing the need for more research on the prevalence of Campylobacter, Salmonella, and Shigella in diarrheal syndromes, particularly in Central Africa. The introduction could benefit from a clearer structure and organization. The text jumps between different topics, such as the global burden of food-borne and water-borne diseases, the prevalence of Campylobacter and Salmonella, and the need for more research in Central Africa. Some of the sentences are quite long and could be broken up for better clarity.

Method: Methodology section is clear and provide a good overview of the systematic review process.  However, including the details of the papers included, the number of papers used from each available source, and the rationale behind the selection of databases are crucial aspects of a systematic review. Additionally, providing more description about the systematic approach used and the methods employed can enhance the transparency and credibility of the review. search strategy is described briefly, but it would be beneficial to provide more information on the specific search terms used and the databases searched.  The inclusion and exclusion criteria are listed, but it would be helpful to provide more context on why certain studies were excluded (e.g., posters, presentations, theses).  The data extraction process is described briefly, more information on how the data was extracted, including any specific tools or software used make the author more confidence about the outcome. The data analysis section needs more elaborate information as more information on the specific statistical methods used, including any assumptions made and how the data was analyzed.

Table 1, 2 and 3- The tables you provided do have a lot of zeros, which could make it challenging to extract meaningful information immediately. You could use a stacked bar chart where each bar represents a country, and segments within the bar show prevalence over the years. This way, the non-zero values will stand out more clearly. Also consider including a heat map instead of the given table.

Discussion : Your  discussion mainly reiterates the findings from individual papers consider having deeper analysis than a summary.

·      Discuss how the studies complement or contradict each other and what the overall trends indicate about the prevalence and impact of these pathogens.

·       Critically evaluate the studies for methodological soundness. Are there any common limitations across studies? How do these limitations affect the interpretation of the results?

·      Place the findings within the broader context of diarrheal disease research in Central Africa and globally. Discuss whether the trends you’ve identified are unique to Central Africa or if they mirror global trends.

·      Identify areas where the research is lacking. For example, are there particular regions in Central Africa that are under-researched? Are there certain populations that have not been adequately studied?

·      Be explicit about the limitations of the review, such as data gaps and inconsistencies in diagnostic methods. Providing recommendations for future research to address these limitations can strengthen the manuscript.

·      The discussion on co-infections versus mono-infections is important. Consider expanding on the implications of these findings and how they impact public health strategies.

·      The discussion on diagnostic methods could benefit from a clearer comparison between conventional methods and molecular techniques. Emphasize how each method impacts the accuracy of pathogen detection.

·      While comparisons with Europe are useful, ensure that these comparisons are balanced and relevant. There are overly detailed descriptions of European surveillance systems.

The study has identified gaps in existing research, but the need for further investigation has not been sufficiently justified. To understand the significance of the research objectives, a more thorough analysis of the data is required. For example, if diarrhea caused by these bacterial species is frequent and severe, it should be investigated more deeply. While cases in Europe are discussed, conditions in Africa may differ due to variations in immune responses. Although prevalence data may be available, it might not have been analyzed in studies, which is a critical issue that needs addressing. Additionally, health system data should be considered in research studies. The review should include specific information on these regional differences and address any studies related to antibiotic resistance in these bacteria.

Author Response

1-for the readability and quick transfer of information, this table can be sepated into two or three tables, taking factors into consideration for ore readability (table 6).

Answer: Table 6, lines 217 to 218, presents all the data used for this review. It could be split into two or three parts. However, this would increase the number of tables and would not allow an overall view or a better appreciation of the data presented.

2-this column should lable properly, the source for the Journal or some other relevent heading (table 6).

Answer: Correct in the manuscript.

3- what do you mean by this, rewrite for the clarity (line 228).

Answer: Correct in the manuscript.

4-Arn't these inofrmation avilable in the national health data bases (line 324).

Answer: These are not standard health databases, but specific health databases and monitoring databases for Campylobacter, Salmonella and Shigella in the sub-region, allowing better visualization and monitoring of the evolution of these pathogens. In fact, data is lacking for certain health databases in Africa, and even when it is provided, it is neglected due to a lack of in-depth segmentation.

Feedback

1-A : Title: My suggestion for the title is “Impact of Campylobacter and Salmonella-Shigella in Diarrheal Infections in Central Africa (1998-2022).

Answer: Correct in the manuscript.

2-A : Abstract: The abstract lacks specific details about the analysis performed on the selected studies. Consider adding a sentence or two to describe the inclusion and exclusion criteria used to select the studies. Provide more detail about the diagnostic tools used in the selected studies. For example, what types of PCR tests were used, and what were the sensitivities and specificities of these tests? The presentation of the data could be improved. The conclusion is quite brief and doesn't provide much insight into the implications of the study's findings. Consider expanding on the conclusion to discuss the significance of the study's results and potential future directions.  Consider including a statement about the limitations of the study, such as the potential for bias in the selected studies or the lack of data on certain aspects of the topic.

Answer: We remain particularly receptive to your request, but we are limited by the word count in the Abstract section, which is 250 words for this journal, and we are already at 247 words. However, you will find the answers to your questions concerning the inclusion and exclusion criteria in the manuscript (line 132-140). As far as PCR is concerned, we're talking about conventional PCR and Real-Time PCR, with some particularities concerning the determination of bacterial genus with either 16S or 23S rRNA. On the other hand, the use of specific genes such as hyppo, GlyA or MapA was not recorded in order to determine the bacterial species. In conclusion, the sheer volume of data recorded is indeed one of the main limitations of the study, allowing us to glimpse and draw attention to these bacterial agents. It is therefore recommended to improve their identification and systematically search for them during diarrhoeal infections.

3-Introduction: Overall, the introduction provides a good overview of the importance of food-borne and water-borne diseases, highlighting the significant morbidity and mortality they cause worldwide. The text also effectively sets the stage for the systematic review by emphasizing the need for more research on the prevalence of Campylobacter, Salmonella, and Shigella in diarrheal syndromes, particularly in Central Africa. The introduction could benefit from a clearer structure and organization. The text jumps between different topics, such as the global burden of food-borne and water-borne diseases, the prevalence of Campylobacter and Salmonella, and the need for more research in Central Africa. Some of the sentences are quite long and could be broken up for better clarity.

Answer: Food- and water-borne diseases are major public health problems worldwide, with significant morbidity and mortality affecting both developed and developing countries [1]. Nearly 600 million people worldwide have been affected by food-related diseases [(3)] [3]. These figures are constantly rising, and the health news linked to the COVID-19 pandemic recently revealed that 3 billion people in Asia and Africa do not have access to a balanced diet [4]. Clearly, these diseases are favored by poor hygiene and difficult access to drinking water sources. Indeed, 28% of the population of sub-Saharan Africa defecates in the open air, and 23% use sanitary facilities that do not ensure hygienic separation of excreta from the immediate human environment [5]. Also, in 2015, nearly 2.5 million people worldwide did not have access to a clean, reliable water source or adequate sanitation facilities [2]. 2.5 billion people still lack access to adequate sanitation and clean water.

Correct in the manuscript.

4-Method: Methodology section is clear and provide a good overview of the systematic review process.  However, including the details of the papers included, the number of papers used from each available source, and the rationale behind the selection of databases are crucial aspects of a systematic review. Additionally, providing more description about the systematic approach used and the methods employed can enhance the transparency and credibility of the review. search strategy is described briefly, but it would be beneficial to provide more information on the specific search terms used and the databases searched.  The inclusion and exclusion criteria are listed, but it would be helpful to provide more context on why certain studies were excluded (e.g., posters, presentations, theses).  The data extraction process is described briefly, more information on how the data was extracted, including any specific tools or software used make the author more confidence about the outcome. The data analysis section needs more elaborate information as more information on the specific statistical methods used, including any assumptions made and how the data was analyzed.

Table 1, 2 and 3- The tables you provided do have a lot of zeros, which could make it challenging to extract meaningful information immediately. You could use a stacked bar chart where each bar represents a country, and segments within the bar show prevalence over the years. This way, the non-zero values will stand out more clearly. Also consider including a heat map instead of the given table.

Answer: -details of the articles included, the number of articles used from each available source: figure 1 and the inclusion and exclusion criteria for our study (line 132-140)

-The rationale for the selection of databases are crucial aspects of a systematic review: to broaden searches and to compensate for the fact that many African publications do not necessarily appear in highly ranked databases such as Medline, Elsevier and others.

-To broaden the search and make up for the fact that many African publications do not necessarily appear on this database: line 128 to 130

The data extraction process is described briefly; more information on how the data were extracted, including the specific tools or software used, allows the author to be more confident about the result: The data were collected daily by 4 of the authors, with retroactive control from one to the other, then double-checked by two supervisors at the end of the chains. Finally, those that were retained were entered into Mendeley to check for duplicates, and the same process was used for the whole section.

-The section on data analysis needs further elaboration: We did not perform statistical tests in this manuscript because the objective was to identify pre-existing data on diarrhoeal infections with Campylobacter, Salmonella and Shigella in the countries of the Central African region between 1998 and 2022. For this purpose, the data were not processed in any way other than the presentation of raw data by year and by country. This presentation allows us to make a better assessment. What you are suggesting would mean changing the whole document.

5-Discussion : Your discussion mainly reiterates the findings from individual papers consider having deeper analysis than a summary.

-Discuss how the studies complement or contradict each other and what the overall trends indicate about the prevalence and impact of these pathogens.

Answer: The aim of this review is to identify studies on the prevalence of Campylobacter, Salmonella and Shigella diarrheal infections in Central African countries. We therefore did not carry out meta-analyses to establish an overall profile of each bacterial agent by country after 25 years. Consequently, we do not have global country trends in our manuscript, but rather country and year trends that are sequential and discontinuous. We cannot compare these trends with those of European, Asian and even West African countries, where data monitoring and surveillance of these pathogens is rigorous and continuous.

Also, no intrinsic comparison can be made between the six countries recorded, as the years of study and profiles are totally different.

-Critically evaluate the studies for methodological soundness. Are there any common limitations across studies? How do these limitations affect the interpretation of the results?

Answer : The methodologies used in all the studies are highly relevant, although repetitive, mainly due to the use of conventional cultivation techniques. However, they all share a common limitation, which is the frequent use of conventional culture methods, which are not very sensitive, not very specific, time-consuming and involve a non-simultaneous search for co-infection. These limitations also affect the interpretation of our results, as they open up few possibilities with regard to the microbial profile of diarrheal infections in Central Africa and the appropriate therapeutic response when treating co-infections.

-Place the findings within the broader context of diarrheal disease research in Central Africa and globally. Discuss whether the trends you’ve identified are unique to Central Africa or if they mirror global trends.

Answer: Yes, they are specific to Central Africa, and they differ from Europe where there is a mass of data. That's why our study focused on Africa, to which our country Gabon belongs, but which also has its own particularities. We wondered whether these germs were involved in diarrhea in Central Africa. We haven't obtained any important data to be able to better appreciate their impact. This lack of data is due to a lack of surveillance programs, a lack of measures to combat these pathogens, outdated, ineffective and time-consuming diagnostic tools, very little use of molecular diagnostics and virtually non-existent technical facilities.

-Identify areas where the research is lacking. For example, are there particular regions in Central Africa that are under-researched? Are there certain populations that have not been adequately studied?

Answer: All our studies point to the fact that research is insufficient, not non-existent, because although it is present in a number of fields, it only covers a very specific area of pathology. The immunological aspect, for example, is little-known.

-Be explicit about the limitations of the review, such as data gaps and inconsistencies in diagnostic methods. Providing recommendations for future research to address these limitations can strengthen the manuscript.

Answer: line 381-389

-The discussion on co-infections versus mono-infections is important. Consider expanding on the implications of these findings and how they impact public health strategies.

Answer: This was mentioned in the manuscript, in Table 5 (lines 256-259) and in the discussion section (lines 291-306). We set out to demonstrate that co-infections are in the minority compared to mono-infections. The data obtained could be explained by the predominance of conventional culture methods, which are time-consuming and affect co-detection capacity.

The impact of this finding on public health strategy would be, firstly, a bias in the diagnosis, monitoring and therapy of diarrhoeal infections. Numerous studies have already pointed to a poly-microbial, bacterial, parasitic and viral mix in diarrhoeal infections. Consequently, the introduction of more robust and effective diagnostic tools for the simultaneous detection of these pathogens via molecular biology would be judicious, and would enable therapeutic regimens to be modified and adapted in the event of co-infections (Part of the discussion).

-The discussion on diagnostic methods could benefit from a clearer comparison between conventional methods and molecular techniques. Emphasize how each method impacts the accuracy of pathogen detection.

Answer: The general idea in the discussion (line 353-363) of the manuscript is to point out that conventional methods are the most widely used for diagnosing these pathogens of interest in Central Africa. These methods are, however, more time-consuming, less sensitive and less specific than molecular methods, which are more sensitive and specific, rapid but more costly. Nevertheless, these conventional methods are important, as they enable a better assessment of the antibiotic susceptibility profile of bacterial agents.

-While comparisons with Europe are useful, ensure that these comparisons are balanced and relevant. There are overly detailed descriptions of European surveillance systems.

Answer: This comparison is not feasible in view of the different means put in place by the European health and epidemiological surveillance system for the control of water- and food-borne toxi-infections due to Campylobacter and Salmonella-Shigella compared with the countries of Central Africa.

The study has identified gaps in existing research, but the need for further investigation has not been sufficiently justified. To understand the significance of the research objectives, a more thorough analysis of the data is required. For example, if diarrhea caused by these bacterial species is frequent and severe, it should be investigated more deeply. While cases in Europe are discussed, conditions in Africa may differ due to variations in immune responses. Although prevalence data may be available, it might not have been analyzed in studies, which is a critical issue that needs addressing. Additionally, health system data should be considered in research studies. The review should include specific information on these regional differences and address any studies related to antibiotic resistance in these bacteria.

Round 2

Reviewer 2 Report

Comments and Suggestions for Authors

Dear Authors,

Your revised manuscript has been carefully revised,

Thanks to your modifications you made, the article is more suitable for publication in its present form,

BR,

Author Response

Here is the reviewer's appreciation: thanks to your modifications you made, the article is more suitable for publication in its present form. He did not give any recommendations for the 2nd round.

Reviewer 3 Report

Comments and Suggestions for Authors

Thank you for addressing the review comments 

Comments on the Quality of English Language

Minor editing is required 

Author Response

He did not give any recommendations for the 2nd round.

Reviewer 4 Report

Comments and Suggestions for Authors

The authors have addressed certain concerns raised during the first revision. They have answered all the queries, though some of the information provided could have been included in the manuscript to improve its quality. However, the present version is acceptable. The abstract requires further improvements for better readability.

Avoid phrases like "over a 25-year period," since you've already mentioned the time range (1998–2022). Just state "from 1998 to 2022."

"Seventeen articles selected, 16 human study and one to animal" is unclear. Better to  revised as "Seventeen articles were selected, including 16 on humans and one on animals."

The conclusion is somewhat weak and repetitive. It can be expanded to highlight key findings more clearly and include actionable recommendations. Instead of just stating that data is lacking, mention what future studies or actions are needed to address the data gaps, especially for Campylobacter in Central Africa.

Author Response

For the part of the abstract :

1-Avoid phrases like "over a 25-year period," since you've already mentioned the time range (1998–2022). Just state "from 1998 to 2022."

Answer: Correct in the manuscript.

2-"Seventeen articles selected, 16 human study and one to animal" is unclear. Better to revised as "Seventeen articles were selected, including 16 on humans and one on animals."

Answer: Correct in the manuscript.

3-The conclusion is somewhat weak and repetitive. It can be expanded to highlight key findings more clearly and include actionable recommendations. Instead of just stating that data is lacking, mention what future studies or actions are needed to address the data gaps, especially for Campylobacter in Central Africa.

Answer: Correct in the manuscript.

Despite the paucity of recorded data on the prevalence of diarrheal infections due to Campylobacter in this sub-region, it is crucial that scientific studies focus on the diagnosis and monitoring of this zoonotic bacterium. Also, improved diagnosis will necessarily involve the integration of molecular tools in the diagnosis of these diarrheic syndromes in both humans and animals.